# Krypton-85 chronometry of spent nuclear fuel

Greg Balco[1], Andrew J. Conant[2], Dallas D. Reilly[3], Dallin Barton[3], Chelsea D. Willett[1], and Brett H. Isselhardt[1]

[1]Lawrence Livermore National Laboratory, 7000 East Ave, Livermore CA USA
[2]Oak Ridge National Laboratory, Oak Ridge TN USA
[3]Pacific Northwest National Laboratory, Richland WA USA
**Correspondence:** Greg Balco (balco1@llnl.gov)

**Abstract.** We describe the use of the radionuclide $^{85}$Kr, which is produced by nuclear fission and has a half-life of 10.76 years, to determine the age of spent nuclear fuel. The method is based on mass-spectrometric measurement of the relative abundance of fissiogenic Kr isotopes extracted from a fuel sample, and we show that it can be applied to micron-scale particles of spent fuel that are analogous to particles that have been released into the environment from various
nuclear facilities in the past. $^{85}$Kr chronometry is potentially valuable for identifying and attributing nuclear materials, grouping samples into collections of common origin that can be used to reconstruct the origin and irradiation history of the material, and verifying declared nuclear activities in the context of international monitoring programs.

## 1 Introduction

In this paper we describe the use of the radionuclide krypton-85, which is produced by uranium and plutonium fission and
has a half-life of 10.76 years (ENDF/B-VIII.0; Brown et al., 2018), as a means of determining the age of spent nuclear fuel. The noble gases Kr and Xe are produced in large quantities by nuclear fission of U and Pu, comprising about 15% of all fission products. Because multiple fissiogenic isotopes of both elements provide a variety of diagnostic isotope ratios, and because their chemically inert nature after production facilitates transport and collection, noble gases produced and released by nuclear activities have been widely proposed and used as detection and monitoring tools. For example,
isotope ratios of Xe and Kr released when reprocessing nuclear fuel have been used for verifying that fuel history, usage, and plutonium production are consistent with declared nuclear activities (Hudson, 1993; Okano et al., 2006). In contrast, this paper focuses on Kr and Xe that have not been released to the atmosphere but are retained within irradiated nuclear fuels and therefore can be used to diagnose the origin, composition, and irradiation history of the material. In general, there exist a wide variety of isotope ratio signatures derived from actinides, decay products, and fission products that can
be used for 'nuclear forensics,' which is concerned with identifying and/or attributing nuclear materials of unknown origin (Moody et al., 2005; Fedchenko, 2015; Kristo et al., 2016). Some of these signatures are radiochronometers that provide information about the date that the material was produced, irradiated, and/or stored, and these are potentially valuable for investigative or forensic applications because they can be used to (i) identify or exclude potential sources of the material; (ii) provide information as to whether material found in the environment was derived from an active nuclear facility or

from long-term storage; and (iii) provide a so-called 'position-independent signature' (e.g., Robel et al., 2018) that can be used to group samples that may have originated from different locations in the same fuel element or assembly. By grouping samples that were irradiated together, variations in irradiation conditions and history inferred from an array of coeval samples can be used to reconstruct reactor type and operating conditions (Dayman and Weber, 2018; Dayman et al., 2019; Savina et al., 2023). Furthermore, the ability to group samples into collections of common origin is particularly important when working with fragments or particles, as might be available in a forensic investigation or an environmental contamination incident, rather than bulk quantities of fuel applicable in routine monitoring of reprocessing facilities.

Because its half-life is appropriate for events postdating the worldwide development of nuclear power in the 1950's, $^{85}$Kr has been proposed as a useful chronometer for spent nuclear fuel (Okano et al., 2006; Park et al., 2010). It has several potential advantages for this purpose, including the relative simplicity of noble gas measurements, the routine availability of noble gas mass spectrometry systems, and the fact that noble gases can typically be extracted from solid samples by heating alone, potentially leaving the sample otherwise undamaged for further analysis. Here we (i) describe the theoretical basis of computing a $^{85}$Kr age from measurements of Kr (and Xe) isotope ratios; (ii) apply the method to bulk and particulate samples from two example spent fuel rods; and (iii) discuss the relative merits of $^{85}$Kr and other isotope ratio chronometers as dating methods and as means of grouping samples.

## 2 Theoretical basis of $^{85}$Kr chronometry

Kr and Xe isotopes produced by nuclear fission include the radioisotope $^{85}$Kr, three stable isotopes of Kr (83,84,86), four stable isotopes of Xe (131,132,134,136), and two short-lived Xe radioisotopes (133, with a half-life of 5 days, and 135, with a half-life of 9 hours). Fissiogenic Kr and Xe isotope ratios are variable among different fissioning nuclides ($^{235}$U, $^{238}$U, $^{239}$Pu) and to a lesser extent with the energy spectrum of neutrons inducing fission (Table 1; Figure 1). Being inert noble gases, Kr and Xe are somewhat mobile within fuel by thermally activated diffusion, and during fuel irradiation are known to both migrate from the fuel matrix into bubbles at grain boundaries and, to some extent, to escape entirely from fuel pellets. Commonly, 10-15% of total Kr and Xe produced during fuel irradiation is found to have been lost from fuel pellets into the plenum of spent fuel rods (Rest et al., 2019). Noble gas mobility, therefore, precludes computing a $^{85}$Kr age simply by measuring the ratio of $^{85}$Kr to its relatively immobile decay product $^{85}$Rb. Instead, it is necessary to measure the ratio of $^{85}$Kr to one or more stable isotopes of Kr that are simultaneously produced and transported. Comparison of the measured isotope ratio to the initial ratio at the time of fission production provides an age.

There are two challenges in formulating a ratio of $^{85}$Kr to other Kr isotopes for use in chronometry. First, after fissiogenic production, the abundances of some stable isotopes of Kr change due to neutron capture. In particular, conversion of $^{83}$Kr to $^{84}$Kr by neutron capture has a relatively high cross-section (Table 1), so the $^{84}$Kr/$^{83}$Kr ratio increases significantly with the total neutron fluence experienced during irradiation, and $^{85}$Kr/$^{83}$Kr or $^{85}$Kr/$^{84}$Kr ratios would therefore have a strong dependence on neutron fluence as well as material age and would not be useful for chronometry. Instead, we

use the ratio $^{85}$Kr/($^{83}$Kr+$^{84}$Kr), which is insensitive to neutron capture after production. Potentially, the ratios $^{85}$Kr/$^{86}$Kr or $^{85}$Kr/($^{83}$Kr+$^{84}$Kr+$^{86}$Kr) could also be used in a similar way (Table 1), but for simplicity we discuss only $^{85}$Kr/($^{83}$Kr+$^{84}$Kr).

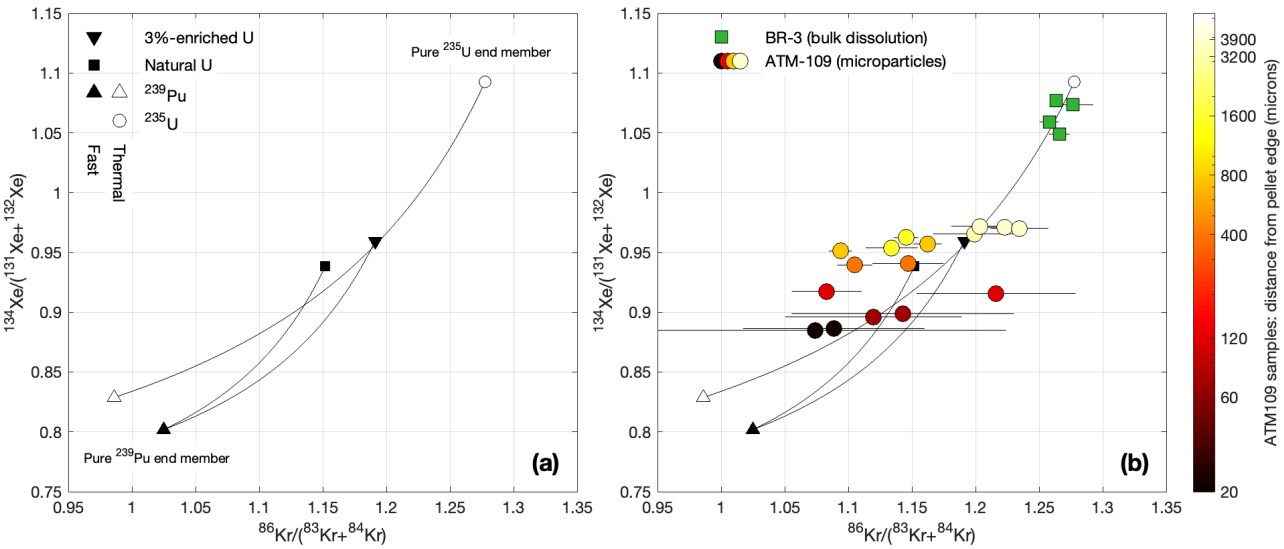

**Figure 1.** Variation in stable isotope ratios of fissiogenic Kr and Xe with fission source. The ratios $^{86}$Kr/($^{83}$Kr+$^{84}$Kr) and $^{134}$Xe/($^{131}$Xe+$^{132}$Xe) are minimally influenced by neutron capture reactions after production and are therefore diagnostic of the fission reactions from which the gas was sourced. Panel (a) shows the expected isotope ratios calculated from cumulative fission yields for end-member fuel compositions and neutron spectra, as well as mixing lines between the end member compositions (see Cassata et al., 2023, for additional details on this diagram). The mixing lines are curved because fission yields are higher for Xe than for Kr. In other words, they represent mixtures of sources rather than mixtures of products: gas compositions produced over a long period of time from an evolving fuel composition could lie anywhere in the region spanned by the curves. Overall, possible fission gas compositions form an array between $^{235}$U and $^{239}$Pu end members, with secondary variability associated with neutron spectrum and contributions from fast neutron fission of $^{238}$U. Panel (b) shows the same predicted array with measured Xe and Kr isotope compositions in the samples analyzed in this study, which span a range of fission source from nearly pure $^{235}$U (for the 8%-enriched, low-burnup BR3 fuel) to a significant fraction of $^{239}$Pu (for the lower-enriched, high-burnup ATM-109 fuel). The results for the BR3 sample are from Cassata et al. (2023).

Second, the initial $^{85}$Kr/($^{83}$Kr+$^{84}$Kr) ratio at the time of Kr production by fission, as noted above, varies among different
fission sources. The fuel in typical nuclear reactors is low-enriched uranium oxide (UO$_2$) with 5% or less $^{235}$U enrichment. When irradiation of fresh fuel begins, production is dominated by $^{235}$U fission. As irradiation proceeds and $^{235}$U is consumed, fission of $^{239}$Pu produced by neutron capture on $^{238}$U accounts for an increasing proportion of fission production of Kr. At high levels of fuel burnup ('burnup' is the cumulative fission energy production per unit mass of nuclear fuel), $^{235}$U is exhausted and Kr production is dominated by $^{239}$Pu fission, with a small contribution from fast neutron-induced
fission of $^{238}$U. Thus, as irradiation proceeds, the initial $^{85}$Kr/($^{83}$Kr+$^{84}$Kr) value of newly produced Kr evolves from a value

characteristic of $^{235}$U fission to a $\sim$15% lower value characteristic of $^{239}$Pu fission. However, it is possible to estimate the initial $^{85}$Kr/($^{83}$Kr+$^{84}$Kr) ratio applicable to Kr extracted from fuel by reference to another Kr isotope ratio, specifically $^{86}$Kr/($^{83}$Kr+$^{84}$Kr), which is also diagnostic of the fission source, also insensitive to neutron capture after production, and includes only stable isotopes so is not modified by radioactive decay after irradiation (Cassata et al., 2023, Figs 1,2).

This strategy is important because it allows a $^{85}$Kr age to be computed from a single measurement of the Kr isotope composition of a gas sample: even though the initial $^{85}$Kr/($^{83}$Kr+$^{84}$Kr) ratio varies with the relative contribution of different fissionable nuclides, it is possible to estimate it without independent knowledge of U and Pu isotope concentrations by exploiting its correlation with a stable isotope ratio that is diagnostic of the fission source.

Thus, our overall procedure for computing a $^{85}$Kr age is to (i) use the observed $^{86}$Kr/($^{83}$Kr+$^{84}$Kr) to estimate the
composition of fissioning actinides responsible for Kr production and therefore the initial $^{85}$Kr/($^{83}$Kr+$^{84}$Kr) ratio, and (ii) compare measured and initial $^{85}$Kr/($^{83}$Kr+$^{84}$Kr) to obtain an apparent gas age. Alternatively, Xe and Kr isotopes are commonly measured simultaneously, so the initial $^{85}$Kr/($^{83}$Kr+$^{84}$Kr) ratio can also be estimated from an equivalent source-sensitive, non-neutron-capture-sensitive ratio of stable Xe isotopes, specifically $^{134}$Xe/($^{131}$Xe+$^{132}$Xe) (Cassata et al., 2023, Figs. 1, 2). This is potentially advantageous because Xe is produced by fission in much greater abundance than
Kr, so Xe isotope ratios are often measured more precisely.

Although it is evident from Figure 2 that knowing either the $^{86}$Kr/($^{83}$Kr+$^{84}$Kr) or $^{134}$Xe/($^{131}$Xe+$^{132}$Xe) ratio provides a fairly close estimate of the initial $^{85}$Kr/($^{83}$Kr+$^{84}$Kr), there is some variability associated with the neutron spectrum responsible for inducing fission. However, this variability is small for typical power reactors, in which fission is dominantly from thermal neutrons and therefore the initial ratios are fairly well approximated by the yields for thermal-neutron-induced fis-
sion shown in Figs 1 and 2. Thus, we use this scenario henceforth to estimate initial $^{85}$Kr/($^{83}$Kr+$^{84}$Kr) ratios. Note that this would not be appropriate for unusual reactor types having an unmoderated fast neutron spectrum; initial $^{85}$Kr/($^{83}$Kr+$^{84}$Kr) ratios from entirely fast-neutron-induced fission would be lower (Fig. 2). However, an unusual reactor type could easily be identified from other isotope ratio signatures.

Given the cumulative fission yields shown in Table 1 and production assumed to be entirely from thermal-neutron-
induced fission, the initial $^{85}$Kr/($^{83}$Kr+$^{84}$Kr) ratio $R_{85,i}$ can be estimated from the measured $^{86}$Kr/($^{83}$Kr+$^{84}$Kr) ratio $R_{86,m}$ by:

$$R_{85,i} = 0.08809 R_{86,m} + 0.07145 \tag{1}$$

or from the measured $^{134}$Xe/($^{131}$Xe+$^{132}$Xe) ratio $R_{134,m}$ by a polynomial fit to the mixing curve shown in Figure 2:

$$R_{85,i} = 1.183 R_{134,m}^3 - 3.734 R_{134,m}^2 + 3.972 R_{134,m} - 1.243 \tag{2}$$

The apparent $^{85}$Kr age $t_{85}$ is then:

$$t_{85} = \frac{-1}{\lambda_{85}} \ln\left(\frac{R_{85,m}}{R_{85,i}}\right) \tag{3}$$

where $R_{85,m}$ is the measured $^{85}$Kr/($^{83}$Kr+$^{84}$Kr) ratio and $\lambda_{85}$ is the $^{85}$Kr decay constant (yr$^{-1}$).

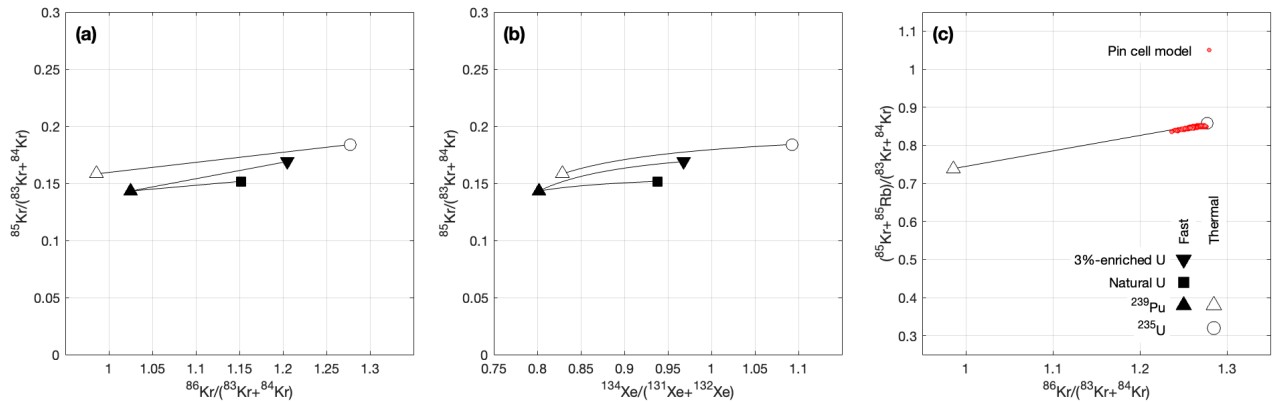

**Figure 2.** Panels (a) and (b) show the variation in the initial $^{85}$Kr/($^{83}$Kr+$^{84}$Kr) ratio at production across a range of fission sources. As in Figure 1, these are calculated for cumulative fission yields for end member fuel compositions and neutron spectra; the symbols are the same in both figures. This ratio is correlated with the stable isotope ratios $^{86}$Kr/($^{83}$Kr+$^{84}$Kr) and $^{134}$Xe/($^{131}$Xe+$^{132}$Xe), so even without independent knowledge of the fuel composition, it is possible to estimate the initial $^{85}$Kr/($^{83}$Kr+$^{84}$Kr) ratio from a measurement of one (or both) of the stable isotope ratios. Panel (c) compares the isotope ratio correlation predicted from the fission yields with the result of a reactor model simulation intended to simulate the irradiation of the BR3 fuel that includes all neutron capture reactions. We use the sum of $^{85}$Kr and its decay product $^{85}$Rb to allow model-data comparison without correcting for $^{85}$Kr decay (see text for detailed discussion). The model results are within 1% of the prediction from the fission yields for enriched uranium with a typical neutron spectrum, showing that neutron fluence effects on the initial ratio are negligible. Note that the mixing lines for pure fast neutron production are not shown in the right panel.

The form of Equation 3 implicitly assumes that all $^{85}$Kr is produced instantaneously and decays for the same length of time. Of course this is not the case in practice, because reactor fuel elements are typically irradiated for several years. This is a significant fraction of the $^{85}$Kr half-life, so substantial decay of $^{85}$Kr occurs simultaneously with production. Thus, the $^{85}$Kr age defined in Equation 3 is strictly an 'apparent age' that does not correspond to a single point-like event (such as the introduction or removal of fuel from the reactor) but instead is only constrained to lie somewhere within the period of fuel irradiation. As discussed below, given the additional assumptions that the Kr production rate is constant, or varies in a known way, during the irradiation, and that all Kr produced is quantitatively retained, one can calculate the expected $^{85}$Kr age corresponding to a particular irradiation history. In general, however, it is not possible to determine either the start or end of an irradiation from the $^{85}$Kr age alone. In practice, this limitation may not be a significant problem for many applications, because typical irradiation durations for commercial reactor fuel are in the 3-5 year range, which is relatively short compared to the many decades since the initial development of nuclear power. Likewise, the application in which

radiochronometry is used to group samples into collections of common origin and irradiation history does not rely on associating a measured age with a specific event.

A final point is that, although $^{85}Kr/(^{83}Kr+^{84}Kr)$ is relatively insensitive to neutron capture reactions during irradiation because the summation of $^{83}Kr$ and $^{84}Kr$ in the denominator removes fluence dependence related to the dominant neutron capture reaction on $^{83}Kr$, the other Kr isotopes have neutron capture cross-sections that are nonzero (although two orders of magnitude less than $^{83}Kr$; see Table 1), so there is likewise the possibility of a nonzero fluence dependence for $^{85}Kr/(^{83}Kr+^{84}Kr)$. We investigated this using the results of the pin cell model for a pressurized water reactor described in Cassata et al. (2023), which tracks all fission, neutron capture, and decay reactions and therefore should expose any deviation in the initial $^{85}Kr/(^{83}Kr+^{84}Kr)$ ratio from the expected fission yield ratios due to neutron interactions. We show the results of this comparison in Figure 2 (right panel), although we include $^{85}Rb$, the decay product of $^{85}Kr$, in the numerator as well so that decay of $^{85}Kr$ during the model simulation does not obscure any fluence effects. The cross-section for neutron capture on $^{85}Rb$ is greater than that for $^{85}Kr$, so this comparison likely yields an upper limit on the effect of fluence on the $^{85}Kr/(^{83}Kr+^{84}Kr)$ ratio. Regardless, deviations between the model simulation and a simple calculation of the expected initial ratio from the fission yields are less than 1% for the entire model simulation. In practice, the uncertainty in estimating $^{85}Kr/(^{83}Kr+^{84}Kr)$ from $^{86}Kr/(^{83}Kr+^{84}Kr)$ or $^{134}Xe/(^{131}Xe+^{132}Xe)$ will therefore most likely be dominated by measurement uncertainty on the isotope ratios, with uncertainty in the assumed neutron spectrum providing a minor contribution. In the presumably unusual case where the reactor type from which a sample originated is completely unknown, uncertainty in the neutron energy spectrum could yield an additional ~4-5% uncertainty in the initial $^{85}Kr/(^{83}Kr+^{84}Kr)$ ratio estimated from one of the stable isotope ratios.

## 3 Sample acquisition and analysis

We describe noble gas measurements on portions of fuel rods from two reactors. Typical commercial power reactor fuel consists of cylindrical $UO_2$ pellets with diameter ~1 cm that are stacked and encapsulated in airtight metallic cladding to form 0.5-4 meter-long rods. The neutron spectrum and flux within a reactor are spatially variable, so different portions of a fuel rod will evolve differently and therefore have somewhat different fission gas isotope compositions. As discussed in the introduction, we are interested in establishing whether the $^{85}Kr$ age of a fuel element is or is not position-independent, so the purpose of sample selection was to capture both axial and radial variability within fuel rods. In addition, the sample set includes both bulk analysis of complete slices of a fuel rod and analysis of individual microparticles excised from bulk fuel.

The first set of samples are slices taken at different axial positions along a fuel rod from Belgian Reactor No. 3 (henceforth, 'BR3'), a pressurized water reactor in Mol, Belgium. This rod had an initial $^{235}U$ enrichment of 8.26% and underwent two periods of irradiation, between July 1976-April 1978 and June 1979-September 1980. This fuel sample has been characterized in detail for a variety of research purposes (e.g., Hanson and Pollington, 2021; Savina et al., 2021, 2023). Four slices from different axial positions spanning the 1-meter length of the rod were completely dissolved in a sealed system,

and the gas released during dissolution was collected for noble gas separation and analysis. Analytical methods for these samples as well as stable Xe and Kr isotope compositions are described in detail in Cassata et al. (2023) and summarized in Table 2. Average burnup inferred from dissolution and actinide analysis of these samples was also reported by

145 Cassata et al. and ranged from 19-48 GWd/ton. Most aspects of the fission gas isotope composition of these samples were discussed in the Cassata reference; here we add discussion of the [85]Kr results.

The second set of samples are microparticles isolated from a slice of a fuel rod that was irradiated in the Quad Cities Unit 1 reactor, a boiling water reactor located in Cordova, IL, USA. This fuel rod had an initial [235]U enrichment of 3%, included 2 wt. % $Gd_2O_3$, and was irradiated between February 1979-September 1987 and then again from November

1989-September 1992. This irradiation is substantially longer than typical in commercial reactor operations. Portions of this fuel were subsequently made available for study as 'Approved Testing Material 109' (henceforth, 'ATM-109'), which has also been characterized for various research purposes (e.g., Wolf et al., 2005; Buck et al., 2015; Pellegrini et al., 2019; Clark et al., 2020). Analyses of various subsamples of this material indicated radially averaged burnup in the range 61-78 GWd/ton (Wolf et al., 2005). We obtained a cross-sectional slice of the ATM-109 material from near the axial center of the

fuel rod and further sectioned it into a matchstick-shaped segment (Figure 3). A focused ion beam (FIB) system was used to cut and separate roughly cube-shaped samples of fuel 1-4 microns in size ("cubes"; see Figs. 3,4) from various radial positions, which were then attached to molybdenum carrier plates for further handling. As spent nuclear fuel is highly radioactive, the purpose of this procedure is mainly to obtain samples that are small enough that they can be worked with in non-radiological, low-background analytical facilities; in addition, these samples are analogous to microparticles

that have been released to the environment from various nuclear facilities in the past (e.g., Steinhauser, 2018). More information on this sampling procedure is provided in Reilly et al. (2020). We cut and analyzed multiple replicate cubes at each radial position.

Xe and Kr were extracted from FIB "cubes" by heating under vacuum. We placed each molybdenum carrier inside a flattened segment ("packet") of Ta tubing that had previously been annealed under vacuum. The packets were placed

under vacuum and heated with a 150 W, 970 nm diode laser. The laser system is equipped with a coaxial optical pyrometer that permits measurement of the packet temperature during heating, and for a subset of the samples we calibrated the pyrometer to the emissivity of the packets, yielding accurate measurements of true temperatures during heating. After initial experiments on some samples to determine the approximate temperature necessary to release noble gases from the fuel matrix, for most samples we applied a heating schedule consisting of a 90-second preheating step at 600° C to

desorb any contaminant Xe and/or Kr derived from the atmosphere, followed by a series of higher-temperature heating steps starting at 1000°-1200° C and increasing in temperature until the amount of Xe released in each step began to decrease significantly. With the exception of one sample, we did not continue to increase the temperature to the point where zero gas was observed in the final step (one of our objectives was to determine whether or not the samples could be recovered for additional analyses, so we did not heat to temperatures higher than necessary). Thus, we can not verify

complete gas extraction for all samples. Based on the results for the one sample that was heated to exhaustion and the fact that we observed near-zero gas release in the final step for many samples, however, we estimate that >95% of

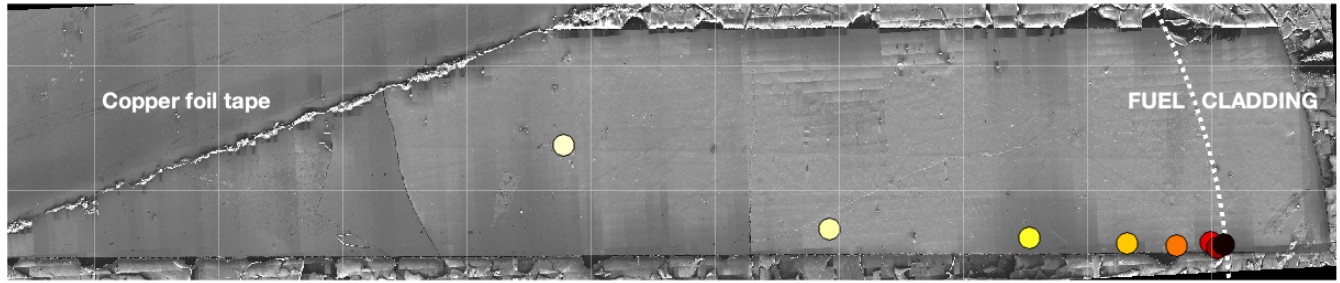

**Figure 3.** Scanning electron microscope image mosaic of slice of ATM-109 fuel pellet sampled for this work. The image shows a matchstick-shaped segment of a disc cut from the middle of a fuel rod, including both fuel and cladding. The surrounding material with a cracked texture visible at the edges of the image is epoxy used for mounting the sample, and part of the sample at upper left has been covered by copper tape. Colored dots show the locations from which samples were removed using a focused ion beam; 2-3 replicate cubes were cut from each location. The same colors are used to indicate radial position in subsequent figures. Light-colored grid lines have 1-millimeter spacing. The diameter of the fuel pellet is 1.06 cm. Images were collected using a FEI Helios NanoLab 660 FIB/SEM at 5 kV and 1.6 nA, and stitched using FEI MAPS 2.5 software.

fission gas present was most likely released from all samples. With the exception of the one sample that was heated to exhaustion at ~1600 °C, we did not exceed 1440 °C in any heating step. These temperatures are well below the melting temperature of uranium oxide (~2800 °C), and microscopic examination of some samples after heating showed that they were intact and could potentially be used for further chemical analysis.

Gas extracted by heating was exposed to hot and cold getters to remove any reactive gases, and Xe and Kr were separated from other noble gases by exposure to a helium-cooled stainless steel cold trap held at 60° K. The trap was then warmed to 220° K and Kr and Xe were let into a Nu Noblesse multicollector noble gas mass spectrometer. All stable isotopes of Xe and Kr, plus $^{85}$Kr, were measured. Detector intercalibration and correction for mass discrimination were accomplished by analysis of an atmospheric Xe-Kr standard that was measured at least twice daily. Background corrections to sample and standard analyses were based on full system "cold" blanks, measured in the same way as the samples except that the packets were not heated. After background correction, further corrections for any atmospheric Xe and/or Kr present were made by assuming that all $^{129}$Xe and $^{80}$Kr observed were atmospheric, and correcting other isotopes accordingly. Thus, data presented in Table 2 are isotope ratios for fissiogenic Xe and Kr after atmospheric correction. "Hot" blanks measured by heating empty Ta packets in the same apparatus released near-background amounts of Xe and Kr with isotopic composition indistinguishable from atmosphere, so correction for any blank contribution from direct or collateral heating of the sample chamber is included in the atmospheric correction and is not considered separately. With one exception, we did not observe any significant variation in Xe or Kr isotope ratios among sequential heating steps for any sample. The exception has to do with a minority of samples that released anomalously large amounts of gas in the 600° preheating step; the isotopic composition of this gas was distinguishable from that of the gas released in subsequent higher-temperature steps. Potentially, this is the result of early release of gas trapped in bubbles that are

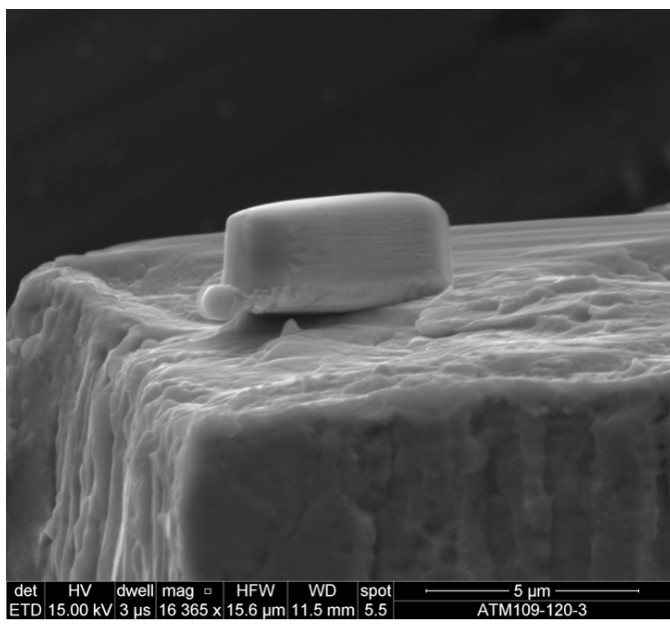

**Figure 4.** Scanning electron microscope image of a sample of ATM-109 fuel prepared for noble gas analysis, collected using a FEI Inspect 4 SEM with the operating parameters shown in the image footer. The sample is a 4-micron-long rectangular solid ("cube") cut from the fuel using a focused ion beam (FIB) system (see Reilly et al., 2020). It is mounted to a molybdenum substrate for handling.

easily decrepitated by heating, but we have not investigated this in detail. Regardless, the observation that there is no significant isotope ratio variation during gas extraction after the preheat step indicates that diffusive fractionation of Xe and Kr isotopes is negligible at experimental heating temperatures, which in turn implies that the completeness of gas extraction is not significant to the interpretation of isotope ratios. Thus, total amounts of all isotopes released in heating steps after the initial preheat step were summed to calculate summary isotope ratios for gas released from each sample. These results are shown in Table 2 and the complete step-degassing data are in the supplement.

## 4   Results and discussion

Figure 1 shows stable Kr and Xe isotope signatures that are diagnostic of fission source. As discussed in Cassata et al. (2023), Kr-Xe data for the BR3 bulk samples indicate that fission gas production is dominated by $^{235}$U fission, as expected for a moderately-enriched fuel that experienced only moderate burnup. Stable isotope ratios for the ATM-109 fuel, on the other hand, indicate a much larger contribution from $^{239}$Pu fission, as expected for a lower-enriched fuel at high burnup that is likely nearly entirely depleted in $^{235}$U. In addition, the ATM-109 data display a strong edge effect such that samples from closer to the edge have a fission gas composition closer to the $^{239}$Pu end member. Again, this is expected from the fact that neutron capture on $^{238}$U and therefore $^{239}$Pu production is highest at the edge of fuel pellets. Overall, the

important observation from these data is that the particulate samples span a much larger range of fission sources and therefore fission gas compositions than the bulk fuel samples.

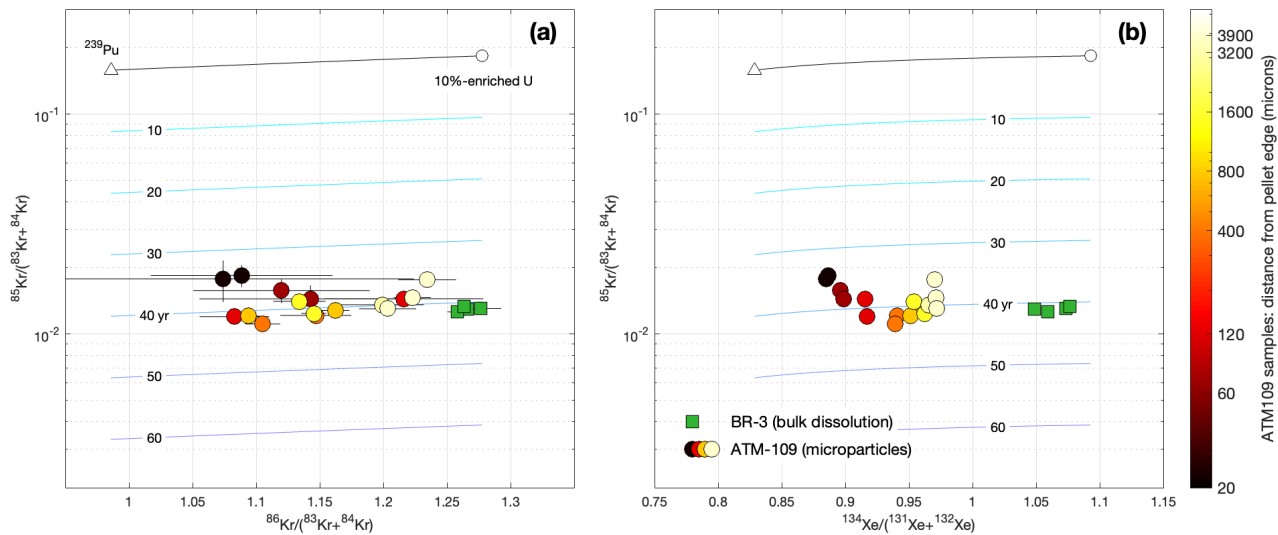

**Figure 5.** Graphical representation of the apparent $^{85}$Kr age of spent fuel samples. The gray line at the top is the initial $^{85}$Kr/($^{83}$Kr+$^{84}$Kr) ratio at production for thermal-neutron-induced fission of mixtures of $^{235}$U and $^{239}$Pu, as shown in Figure 2. As discussed above, the initial ratio is weakly variable among fission source but can be inferred from its correlation with the stable isotope ratios $^{86}$Kr/($^{83}$Kr+$^{84}$Kr) (panel a) or $^{134}$Xe/($^{131}$Xe+$^{132}$Xe) (panel b). Decay of $^{85}$Kr after production then causes the $^{85}$Kr/($^{83}$Kr+$^{84}$Kr) ratio to decrease over time, as indicated by the colored isochrons labeled in years. This allows an apparent age of the fission gas to be calculated. Error bars on data are $1\sigma$ and, where not visible, are smaller than the size of the plotting symbols. Comparison of these plots shows that, for the particulate samples from ATM-109, relatively large uncertainties in the $^{86}$Kr/($^{83}$Kr+$^{84}$Kr) ratio measured on small amounts of gas mean that the initial $^{85}$Kr/($^{83}$Kr+$^{84}$Kr) can be estimated more accurately from the Xe data. Note that there is a 3-year gap between $^{85}$Kr/($^{83}$Kr+$^{84}$Kr) measurements for the two sample sets (Table 3), but in this figure they are plotted as measured at the time of analysis and have not been normalized to a common analysis date.

Figure 5 shows the relationship of the $^{85}$Kr data to the stable Kr and Xe isotope ratios indicative of fission source. Across the full range of stable isotope compositions, the differences between measured $^{85}$Kr/($^{83}$Kr+$^{84}$Kr) ratios and production ratios expected from fission yields are consistent with 34-43 years of decay following irradiation. Apparent ages of the BR3 fuel show minimal axial variation among samples. As observed in the stable isotope ratios, apparent ages of the ATM-109 particle samples show significant radial variability, with samples near the edge having younger apparent ages. It is evident from both Figures 1 and 5 that for the ATM-109 particle samples, Kr isotope ratios have significantly larger uncertainties and scatter than Xe isotope ratios; this is primarily due to low abundance of Kr. For mass-spectrometric measurement of these samples, signals for Kr isotopes on ion counters were in the range 10-100 cps, whereas Xe isotope signals were an

order of magnitude higher. For the bulk dissolution samples, a large quantity of gas was available and Kr measurement precision was not signal-limited.

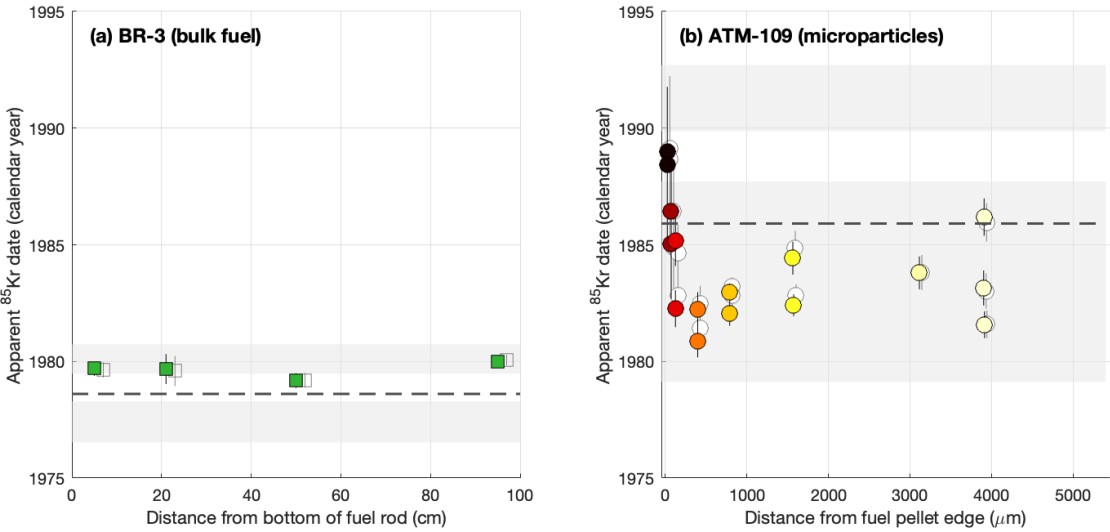

**Figure 6.** Apparent $^{85}$Kr dates for bulk fuel samples from BR3 (a) and microparticle samples from ATM-109 (b). The prominent colored symbols are the dates calculated using the $^{134}$Xe/($^{131}$Xe+$^{132}$Xe) ratio to estimate the initial $^{85}$Kr/($^{83}$Kr+$^{84}$Kr) ratio, and the lighter unshaded symbols behind them are calculated using the $^{86}$Kr/($^{83}$Kr+$^{84}$Kr) ratio as the estimator. The two approaches yield indistinguishable results. The shaded regions are the periods of time during which the respective fuels were irradiated, and the dashed lines are the apparent $^{85}$Kr dates expected given simple assumptions that $^{85}$Kr production is constant throughout the irradiation and all gas is retained. The color-coding of the symbols is the same as in previous figures.

Table 3 shows apparent $^{85}$Kr dates calculated from Kr and Xe isotope ratios, and Figure 6 compares them with the known irradiation dates of the two fuels. Apparent $^{85}$Kr ages in this figure and in Table 3 have been calculated using
Equations 1-3, in which we assume that initial $^{85}$Kr($^{83}$Kr+$^{84}$Kr) ratios are characteristic of thermal-neutron-induced fission as shown in Figs. 2 and 5, and use this mixing curve with the measured $^{86}$Kr/($^{83}$Kr+$^{84}$Kr) or $^{134}$Xe/($^{131}$Xe+$^{132}$Xe) ratios to estimate the initial ratio applicable to each sample. We now discuss several aspects of these results.

First, we discuss internal variation in apparent $^{85}$Kr dates for both fuels. Dates for the BR3 bulk fuel samples are indistinguishable across a range of axial positions. On the other hand, $^{85}$Kr dates on the ATM-109 particle samples show
both (i) variance in excess of measurement uncertainty among replicate samples taken from the same radial position, and (ii) an edge effect in which $^{85}$Kr dates from samples within ∼100 $\mu$m of the fuel pellet edge are significantly younger than those from the pellet center. We hypothesize that the scatter among replicates is likely due to partitioning of fission gas between fuel matrix and bubbles due to gas mobility during irradiation. Xe and Kr are known to migrate into micron- or sub-micron-scale bubbles during fuel irradiation (see, e.g., Rest et al., 2019) and therefore, on average, gas in bubbles

must have an older production age than gas in the matrix. This would imply micron-scale inhomogeneity in gas age, which appears to have been captured by our sampling procedure.

The edge effect is nearly certainly due to gas loss associated with so-called "high-burnup structure" (e.g., Rondinella and Wiss, 2010) at fuel pellet edges. High-burnup structure forms by recrystallization of fuel subjected to high local concentrations of fissions, primarily at the outer surface of fuel pellets. Various microbeam studies of spent fuel (Walker
et al., 2012) have observed lower concentrations of fission gases within regions of high-burnup structure, implying that recrystallization of the fuel matrix either directly rejects noble gases or temporarily or permanently enhances their diffusivity. If the development of high-burnup structure late in the irradiation results in the loss of gas produced early in the irradiation, then only gas produced late in the irradiation remains in the sample for measurement, and this gas will have a younger $^{85}$Kr age than the gas that was lost. Thus, a sample that experienced formation of high-burnup structure late in
the irradiation will have a younger apparent $^{85}$Kr age than an equivalent sample that retained all gas produced during the irradiation. The observation of younger apparent $^{85}$Kr ages in edge samples where high-burnup structure is expected is exactly consistent with this mechanism.

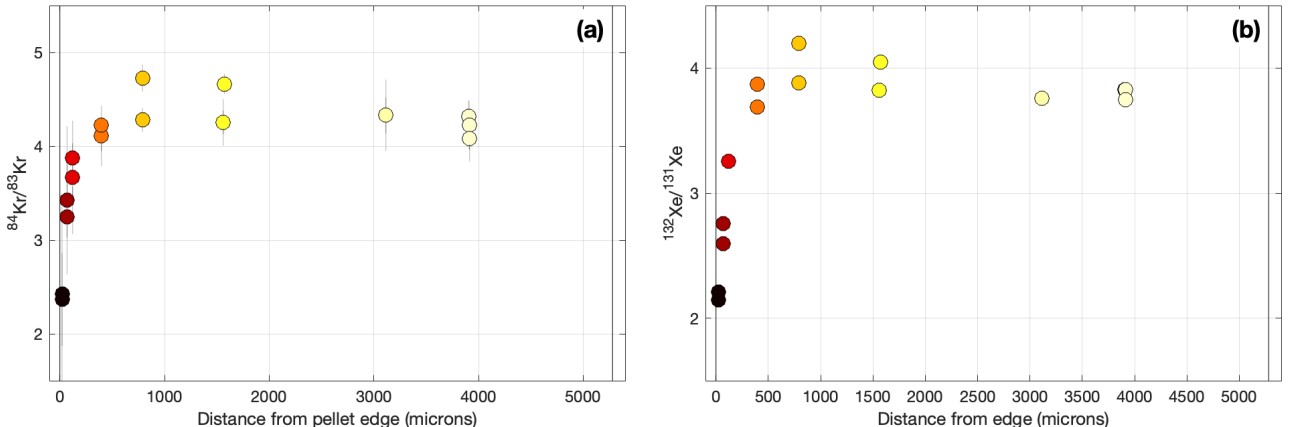

**Figure 7.** Radial variation in fluence-sensitive $^{84}$Kr/$^{83}$Kr (a) and $^{132}$Xe/$^{131}$Xe (b) ratios in ATM-109 fuel. Color-coding of symbols is the same as in previous figures.

The hypothesis that the edge effect in apparent $^{85}$Kr ages in the ATM-109 fuel is due to gas loss is consistent with other stable isotope evidence. Specifically, again, by 'gas loss' we refer to preferential loss of gas produced early in the
250 irradiation, either by a continuous process active throughout the irradiation (e.g., thermally activated diffusion) or a distinct event (e.g., formation of high-burnup structure). As diffusive fractionation of Xe and Kr isotopes in these samples appears to be negligible (see discussion in section 3 above), we have no reason to believe that the process of gas loss itself affects the isotope composition of the gas that is being lost; instead, isotope ratios are different between samples that have and have not experienced gas loss because the gas that is still present and available for measurement was produced during

different parts of the irradiation. For example, Figure 7 shows radial variation in the ratios $^{84}$Kr/$^{83}$Kr and $^{132}$Xe/$^{131}$Xe. Because of the high neutron capture cross-sections of $^{83}$Kr and $^{131}$Xe, these ratios are diagnostic of total neutron fluence experienced by the fuel and should therefore increase monotonically with irradiation time (Cassata et al., 2023). However, measured ratios are much lower near the fuel pellet edge. As the edge and center samples themselves have experienced the same irradiation time, the only explanation for this is that the fission gas present in edge samples has experienced a
shorter irradiation time than that in the pellet center. This is only possible if a significant fraction of gas produced early in the irradiation has been lost from edge samples.

Second, we compare observed to expected $^{85}$Kr ages. Because both $^{85}$Kr production and decay are taking place throughout the period of irradiation, an apparent $^{85}$Kr gas age is expected to date some time in the middle of the irradiation. In Figure 6 we calculate an expected age for each fuel from the known irradiation dates by making the simplifying
assumptions that the Kr production rate is constant throughout the irradiation and all gas is retained. The results differ from this expectation in several ways. With the exception of the edge samples that are biased young by gas loss as discussed above, the ATM-109 particles have apparent ages older than expected. This is relatively easy to explain by the observation that Kr fission yields from $^{239}$Pu are 40-60% lower than from $^{235}$U. For this unusually-high-burnup fuel, the Kr production rate has therefore nearly certainly decreased over time as $^{235}$U is depleted and Kr production late in
the irradiation becomes dominated by $^{239}$Pu fission. If the Kr production rate has decreased throughout the course of the irradiation and all gas is retained, the apparent $^{85}$Kr age will be older than the age expected from a constant-production assumption. This appears to be the case for particle samples from the center of the ATM-109 fuel pellet. Overall, for the ATM-109 particle data, offsets from the $^{85}$Kr age expected from simple assumptions are significant with respect to measurement uncertainties and spatially variable, but can be satisfactorily explained by steadily decreasing Kr production
during a long irradiation as well as gas loss during formation of high-burnup structure at the pellet edge.

On the other hand, apparent $^{85}$Kr dates for the BR3 fuel are younger than expected for constant production, and close to the end of the irradiation. This is unexpected for this 8%-enriched, moderate-burnup fuel, for which the assumption of constant production during the irradiation should be much closer to correct than for the 3%-enriched, high-burnup ATM-109 fuel. Model simulations of the BR3 fuel (Cassata et al., 2023, see also Fig. 2) predict that its apparent age at
discharge should be within 0.05-0.1 yr of the expected age computed from a constant-production assumption. It is likely that a nonzero amount of fission gas was lost from the fuel to the rod plenum during irradiation, and this gas was not captured in the analysis of these samples, so a young bias to the age due to gas loss is possible. However, typical losses of 10-15% are insufficient to explain the magnitude of the observed young bias in apparent ages. In addition, gas loss would be expected to be highest at the axial center of the rod where fuel temperatures are the highest, so if gas loss
was a good explanation for the young bias, we would see younger apparent ages in the axial center of the rod. In fact, we see a slightly older (although indistinguishable at uncertainty) age from the axial center sample. Overall, although the apparent ages from the BR3 samples fall within the known period of irradiation, we do not have a satisfying explanation for why they are systematically 1-2 years younger than expected.

Third, we discuss the implications of age variations for grouping of samples. Although it is evident from the BR3 results that the apparent $^{85}$Kr age is position-independent at the bulk sample level, the ATM-109 microparticle results show that this is not strictly the case at the level of individual particles. All apparent ages from ATM-109 samples fall within the period during which the fuel was irradiated, but variations in apparent age in excess of measurement uncertainty are evident both across the radial array of samples and in replicate data from the same radial position. Although the long irradiation and high burnup of the ATM-109 fuel likely give rise to more internal variability than would be expected for a fuel with a more typical history, strictly, these samples could not be associated with each other purely on the basis of the apparent $^{85}$Kr age alone. However, the apparent $^{85}$Kr age is not determined in isolation, but as part of a larger set of Xe and Kr isotope ratio measurements that provides additional leverage in grouping samples. It is clear from, for example, Figure 5, that even though the ATM-109 samples do not all have exactly the same $^{85}$Kr age, when multiple isotope ratios are considered they belong to a coherent array.

Finally, we highlight several aspects of uncertainty analysis. Given the assumption (discussed above) that the neutron spectrum is typical of a normal commercial power reactor, nominal measurement uncertainty in the apparent $^{85}$Kr age is dominated by measurement uncertainty on $^{85}$Kr and to a lesser extent on other Kr isotopes, especially for signal-limited measurements from small particles. These nominal uncertainties (Table 2) are 0.2-0.6 yr for non-signal-limited measurements from the BR3 dissolution gas and 0.5-3 yr for lower-abundance measurements on the ATM-109 microparticles. As noted above, an additional uncertainty on the initial $^{85}$Kr/($^{83}$Kr+$^{84}$Kr) ratio would arise if the reactor type and therefore the neutron spectrum responsible for fission were completely unknown. However, it is evident from Figure 6 that the most important uncertainty in interpreting apparent $^{85}$Kr ages arises not from measurement uncertainties but from deviations between the observed ages and the ages that we would expect to observe given the known irradiation histories of the samples. These are larger than measurement uncertainty and are related to irradiation history and duration, fuel enrichment and burnup, and, at least in the high-burnup ATM-109 samples, enhanced fission gas loss near fuel pellet edges. These complications could be important if, for example, seeking to use $^{85}$Kr age data to associate fuel samples with one of several candidate irradiation periods, or to infer the irradiation duration of fuel samples with a known discharge date. Overall, systematic variations in apparent $^{85}$Kr age that depend on irradiation conditions are a more important limit to the interpretability of these data than the measurement uncertainties.

## 5   $^{85}$Kr compared to other spent fuel chronometers

Other radiochronometers have been proposed and/or used for spent fuel samples. Their relative applicability and usefulness compared to $^{85}$Kr chronometry are likely to vary with fuel enrichment and burnup, type and amount of sample, and available measurement technology. However, here we highlight some general differences between $^{85}$Kr and two other proposed chronometers that have been applied to small particle samples of spent fuel: $^{90}$Sr (Savina et al., 2023) and $^{241}$Pu (Hanson and Pollington, 2021).

$^{90}$Sr is a fission product with a half-life of 28.2 years that can be used as a chronometer by comparison with the stable fission product $^{88}$Sr. As for the ratio $^{85}$Kr/($^{83}$Kr+$^{84}$Kr), the initial $^{90}$Sr/$^{88}$Sr ratio depends on the proportion of fissions derived from the various actinide isotope and neutron energies. With $^{85}$Kr, we exploit variations in stable Kr and Xe isotopes that are produced simultaneously to infer the initial $^{85}$Kr/($^{83}$Kr+$^{84}$Kr) ratio, and model simulations show that the initial ratio estimate has percent-level uncertainty. For $^{90}$Sr, the $^{90}$Sr/$^{88}$Sr ratio only varies at the percent level across the full range of fission sources, so can be estimated with comparable precision. Savina et al. (2023) used a resonance ionization mass spectrometer (RIMS) to measure $^{90}$Sr/$^{88}$Sr ages on 10-micron microparticles derived from a sample of the BR3 fuel, and obtained apparent ages very close to the age expected from constant production (1-2 years older than our apparent $^{85}$Kr ages for bulk samples of this fuel). Important differences between the $^{85}$Kr and $^{90}$Sr chronometers are as follows:

- $^{85}$Kr has a lower fission yield than $^{90}$Sr and also can be lost by diffusion, so the $^{90}$Sr concentration in spent fuel is expected to be higher. Thus, measurement precision on $^{90}$Sr may be better for small samples.

- The production rate of both $^{90}$Sr and $^{85}$Kr is expected to be relatively constant throughout an irradiation in most cases. Thus, apparent ages for both should fall near the middle of the irradiation period.

- Sr is expected to be far less diffusively mobile than Kr, which implies that variability in apparent ages at the microparticle scale that are caused by Kr mobility should not be evident in $^{90}$Sr ages.

- Measurement of Kr isotopes utilizes the relatively simple and widely used analytical method of heating under vacuum followed by noble gas mass spectrometry, whereas measurement of Sr isotopes requires either sample dissolution and chemical separation, which is complex and time-consuming, or measurement by RIMS, which is not widely available.

$^{241}$Pu is produced by neutron capture during irradiation and decays to $^{241}$Am with a half-life of 14.35 years. Thus, the $^{241}$Am/$^{241}$Pu ratio in spent fuel provides an age estimate for the fuel. This chronometer was, for example, applied to 10- to 20-micron size samples of the BR3 fuel by Hanson and Pollington (2021). By comparison with $^{85}$Kr chronometry,

- Due to neutron capture on $^{241}$Am, the $^{241}$Am/$^{241}$Pu ratio at discharge is lower than expected from $^{241}$Pu production alone, and varies with fluence. Thus, to achieve comparable accuracy to the $^{85}$Kr or $^{90}$Sr chronometers, it would most likely be necessary to apply an independent fluence signature to account for $^{241}$Am loss during irradiation.

- Production of $^{241}$Pu requires three successive neutron capture reactions, so is weighted toward the end of an irradiation. Thus, a $^{241}$Pu age is expected to date a later time in an irradiation than an $^{85}$Kr or $^{90}$Sr age.

- Measurement of the $^{241}$Am/$^{241}$Pu ratio requires sample dissolution and chemical separation of Am from Pu, followed by two separate isotope dilution measurements. Thus, it is much more complex and time-consuming than measurement of Kr isotope ratios by vacuum heating/NGMS or measurement of Sr isotope ratios by RIMS.

– Am and Pu are not expected to be diffusively mobile, so in principle $^{241}$Am/$^{241}$Pu ages should display less internal variability than $^{85}$Kr ages. However, in the one application of this method to small particle samples, Hanson and Pollington (2021) observed relatively high and apparently nonsystematic internal variability, well in excess of nominal measurement uncertainty.

## 6   Conclusions

The date of irradiation of spent reactor fuel can be determined by measuring the $^{85}$Kr/($^{83}$Kr+$^{84}$Kr) ratio of gas extracted from spent fuel samples, and using simultaneous measurements of other Kr and Xe isotope ratios to estimate the initial $^{85}$Kr/($^{83}$Kr+$^{84}$Kr) ratio at the time of fission gas production. This method has several possible advantages in relation to other radiochronometric methods, in particular that it can be applied to micron-size samples of spent fuel and only requires a single simultaneous measurement of Kr and Xe isotopes, which is relatively rapid and utilizes fairly simple and widely available vacuum heating and noble gas mass spectrometry systems. When applied to microparticles of spent U oxide fuel, the method is nondestructive in the sense that the samples are not melted or otherwise damaged and could be subjected to further chemical analysis, although it is possible that some other volatile elements besides the noble gases are depleted during heating.

$^{85}$Kr chronometry applied to two sets of samples of spent reactor fuel yields apparent ages that, as expected, fall entirely within known periods of irradiation. From this perspective, the method is successful. When considered in more detail, we observed both systematic (e.g., from edge to center of a fuel pellet) and random (e.g., variability among replicates) variation in apparent age among microparticle samples, as well as differences between measured $^{85}$Kr ages and expected ages near irradiation midpoints for both bulk and particulate samples. We attribute these observations to aspects of the fuel history (e.g., irradiation duration and burnup) as well as mobility and loss of noble gases within uranium oxide fuel.

From the perspective of using the $^{85}$Kr chronometer to group samples of unknown origin, indistinguishable $^{85}$Kr ages from bulk dissolutions of fuel slices at different axial positions show that the $^{85}$Kr age is a position-independent signature at the bulk sample level. On the other hand, observed variations in apparent $^{85}$Kr ages among microparticles from the same fuel sample show that this is not strictly the case at the microparticle scale. Especially for high-burnup fuel, internal variation in $^{85}$Kr ages at the particle scale is likely. However, both our observations and our explanation of observed internal variability indicate that the magnitude of internal variability (i) cannot exceed the duration of fuel irradiation, and (ii) is most likely maximized in high-burnup samples. Thus, for typical fuel irradiation durations of 1-2 years, internal variability in $^{85}$Kr ages on particle samples will likely not exceed measurement uncertainty, and therefore will not preclude accurate grouping of samples or association of samples with candidate irradiation periods. In addition, when both the $^{85}$Kr age and other simultaneously measured stable Kr and/or Xe isotope ratios are considered, data from particles of common origin will form coherent arrays. Even in the presence of some internal variability in $^{85}$Kr ages, sample grouping can be improved by considering the $^{85}$Kr age as part of a set of multiple noble gas isotope ratios.

*Data availability.* All data described in this paper are included in the tables and the supplement.

*Author contributions.* Conceptualization: GB, AJC, DDR, DB, BHI; Methodology: GB, AJC, DDR, DB; Software: GB, AJC; Validation: GB, AJC; Formal analysis: GB, AJC, CDW; Investigation: GB, CDW; Resources: GB, AJC, CDW; Writing (original draft): GB, CDW; Writing (review and editing): all authors; Visualization: GB, DDR, DB; Project administration: GB, BHI; Funding acquisition: AJC, BHI.

*Competing interests.* Balco is a member of the Geochronology editorial board.

*Acknowledgements.* We thank Autumn Roberts for assistance with electron microscope imaging. Work at LLNL was performed under the auspices of the U.S. Department of Energy under Contract DE-AC52-07NA27344; this is LLNL-JRNL-861096-DRAFT.

**Table 1: Cumulative fission yields, cumulative fission yield ratios, and neutron capture cross-sections for relevant Kr isotopes. Data are from ENDF/B-VIII.**

| | $^{235}U_{therm}$ | $^{235}U_{fast}$ | $^{238}U_{fast}$ | $^{239}Pu_{therm}$ | $^{239}Pu_{fast}$ | Neutron capture x-section s(n,g) (barns) | | |
| | | | | | | Thermal | Resonance | Fast |
| Cumulative fission yields (%) | | | | | | | | |
| $^{82}Se$ | | | | | | <0.1 | 0.2 | <0.1 |
| $^{83}Kr$ | 0.536 | 0.577 | 0.396 | 0.297 | 0.315 | 198 | 188 | <0.1 |
| $^{84}Kr$ | 1.002 | 1.031 | 0.826 | 0.480 | 0.453 | 0.1 | 2.1 | <0.1 |
| $^{85}Kr$ | 0.283 | 0.275 | 0.149 | 0.123 | 0.110 | 1.7 | 2.7 | <0.1 |
| $^{86}Kr$ | 1.965 | 1.947 | 1.296 | 0.766 | 0.787 | <0.1 | <0.1 | <0.1 |
| Cumulative fission yield ratios | | | | | | | | |
| $^{85}Kr/^{86}Kr$ | 0.144 | 0.141 | 0.115 | 0.160 | 0.139 | | | |
| $^{85}Kr/(^{83}Kr+^{84}Kr)$ | 0.184 | 0.171 | 0.122 | 0.158 | 0.143 | | | |
| $^{85}Kr/(^{83}Kr+^{84}Kr+^{86}Kr)$ | 0.081 | 0.077 | 0.059 | 0.080 | 0.071 | | | |

**Table 2. Isotopic composition of fissiogenic Xe and Kr in spent fuel samples.**

| Sample name | | Relative Kr abundances (Kr-86 = 100) | | | | | | Relative Xe abundances (Xe-134 = 100) | | | | | |
|---|---|---|---|---|---|---|---|---|---|---|---|---|---|
| | | Kr-83 | +/- | Kr-84 | +/- | Kr-85 | +/- | Xe-131 | +/- | Xe-132 | +/- | Xe-136 | +/- |
| ATM-109 microparticle samples | | | | | | | | | | | | | |
| | Radial distance from pellet edge (microns) | | | | | | | | | | | | |
| ATM-109-20-54 | 22 | 26.8 | 1.3 | 65.1 | 5.8 | 1.694 | 0.166 | 35.87 | 0.20 | 76.95 | 0.35 | 157.05 | 0.61 |
| ATM-109-20-55 | 22 | 27.7 | 2.9 | 65.4 | 12.5 | 1.656 | 0.285 | 35.20 | 0.33 | 77.81 | 0.51 | 156.76 | 0.81 |
| ATM-109-60-54 | 70 | 21.0 | 1.2 | 68.3 | 5.3 | 1.408 | 0.155 | 31.02 | 0.18 | 80.57 | 0.34 | 156.99 | 0.56 |
| ATM-109-60-55 | 70 | 19.8 | 1.4 | 67.7 | 6.4 | 1.264 | 0.168 | 29.62 | 0.20 | 81.65 | 0.38 | 156.13 | 0.64 |
| ATM-109-120-01 | 122 | 17.6 | 1.0 | 64.6 | 4.0 | 1.189 | 0.063 | 25.69 | 0.17 | 83.54 | 0.36 | 155.55 | 0.81 |
| ATM-109-120-02 | 122 | 18.93 | 0.55 | 73.4 | 2.2 | 1.107 | 0.050 | 25.63 | 0.14 | 83.38 | 0.32 | 154.96 | 0.88 |
| ATM-109-400-01 | 397 | 17.07 | 0.51 | 70.1 | 2.0 | 1.058 | 0.043 | 21.83 | 0.12 | 84.47 | 0.31 | 152.95 | 0.80 |
| ATM-109-400-02 | 397 | 17.33 | 0.29 | 73.2 | 1.0 | 1.004 | 0.043 | 22.71 | 0.11 | 83.74 | 0.31 | 152.56 | 0.86 |
| ATM-109-800-01 | 790 | 15.02 | 0.20 | 71.0 | 0.8 | 1.102 | 0.025 | 20.100 | 0.076 | 84.38 | 0.23 | 151.13 | 0.67 |
| ATM-109-800-02 | 790 | 17.32 | 0.27 | 74.1 | 0.6 | 1.103 | 0.037 | 21.55 | 0.12 | 83.59 | 0.35 | 151.91 | 0.99 |
| ATM-109-1600-01 | 1562 | 16.78 | 0.40 | 71.4 | 1.5 | 1.240 | 0.052 | 21.75 | 0.11 | 83.09 | 0.30 | 152.01 | 0.81 |
| ATM-109-1600-02 | 1572 | 15.42 | 0.21 | 71.90 | 0.61 | 1.083 | 0.032 | 20.59 | 0.10 | 83.32 | 0.31 | 151.73 | 0.94 |
| ATM-109-3200-01 | 3115 | 15.65 | 0.52 | 67.7 | 2.2 | 1.132 | 0.043 | 21.77 | 0.12 | 81.80 | 0.29 | 151.67 | 0.80 |
| ATM-109_3-1 | 3905 | 15.38 | 0.26 | 66.41 | 0.90 | 1.195 | 0.055 | 21.321 | 0.069 | 81.65 | 0.19 | 151.30 | 0.43 |
| ATM-109_3-2 | 3910 | 15.50 | 0.38 | 65.5 | 1.4 | 1.428 | 0.069 | 21.344 | 0.094 | 81.74 | 0.26 | 151.25 | 0.65 |
| ATM-109_3-3 | 3915 | 16.34 | 0.37 | 66.8 | 1.5 | 1.083 | 0.035 | 21.686 | 0.082 | 81.21 | 0.20 | 151.58 | 0.53 |
| BR-3 total dissolutions (reproduced from Cassata et al., 2023) | | | | | | | | | | | | | |
| | Axial distance from rod bottom (cm) | | | | | | | | | | | | |
| G14 | 95 | 25.98 | 0.39 | 52.34 | 0.90 | 1.023 | 0.012 | 34.359 | 0.019 | 58.799 | 0.057 | 115.492 | 0.082 |
| G16 | 5 | 25.62 | 0.31 | 53.38 | 0.40 | 1.026 | 0.041 | 34.84 | 0.12 | 60.50 | 0.12 | 118.33 | 0.31 |
| G18 | 50 | 23.32 | 0.16 | 56.17 | 0.45 | 1.001 | 0.017 | 29.774 | 0.092 | 64.64 | 0.16 | 138.95 | 0.23 |
| G21 | 21 | 24.36 | 0.11 | 54.79 | 0.16 | 1.057 | 0.011 | 30.378 | 0.042 | 62.486 | 0.071 | 134.88 | 0.26 |

Notes:
Relative abundances shown for ATM-109 particles reflect the sum of gas released in all heating steps after the initial preheat.

**Table 3. $^{85}$Kr age calculations.**

| Sample name | Analysis date | Measured $^{85}$Kr/($^{83}$Kr+$^{84}$Kr) | +/- | Measured $^{134}$Xe/($^{131}$Xe+$^{132}$Xe) | +/- | Estimated initial $^{85}$Kr/($^{83}$Kr+$^{84}$Kr) | +/- | $^{85}$Kr age (yr before analysis) | +/- (yr) | $^{85}$Kr age (calendar date) | +/- (days) |
|---|---|---|---|---|---|---|---|---|---|---|---|
| ATM-109 microparticle samples | | | | | | | | | | | |
| ATM-109-20-54 | 4/19/23 | 0.0184 | 0.0021 | 0.8864 | 0.0036 | 0.1689 | 0.0018 | 34.3 | 1.8 | 12/23/88 | 650 |
| ATM-109-20-55 | 4/19/23 | 0.0178 | 0.0038 | 0.8848 | 0.0053 | 0.1687 | 0.0019 | 34.9 | 3.3 | 6/11/88 | 1200 |
| ATM-109-60-54 | 4/19/23 | 0.0158 | 0.0019 | 0.8962 | 0.0034 | 0.1703 | 0.0018 | 36.9 | 1.9 | 6/4/86 | 700 |
| ATM-109-60-55 | 4/19/23 | 0.0144 | 0.0022 | 0.8987 | 0.0039 | 0.1706 | 0.0018 | 38.3 | 2.3 | 1/15/85 | 850 |
| ATM-109-120-01 | 8/10/23 | 0.0145 | 0.0010 | 0.9155 | 0.0038 | 0.1727 | 0.0018 | 38.4 | 1.1 | 3/10/85 | 400 |
| ATM-109-120-02 | 8/11/23 | 0.01199 | 0.00060 | 0.9173 | 0.0035 | 0.1729 | 0.0018 | 41.34 | 0.79 | 4/9/82 | 290 |
| ATM-109-400-01 | 8/12/23 | 0.01214 | 0.00055 | 0.9407 | 0.0034 | 0.1752 | 0.0018 | 41.37 | 0.73 | 3/31/82 | 270 |
| ATM-109-400-02 | 8/13/23 | 0.01110 | 0.00048 | 0.9394 | 0.0034 | 0.1751 | 0.0018 | 42.74 | 0.69 | 11/13/80 | 250 |
| ATM-109-800-01 | 8/14/23 | 0.01281 | 0.00030 | 0.9571 | 0.0026 | 0.1766 | 0.0018 | 40.65 | 0.39 | 12/18/82 | 140 |
| ATM-109-800-02 | 8/15/23 | 0.01206 | 0.00039 | 0.9511 | 0.0039 | 0.1761 | 0.0018 | 41.54 | 0.53 | 1/30/82 | 190 |
| ATM-109-1600-01 | 8/16/23 | 0.01407 | 0.00062 | 0.9539 | 0.0033 | 0.1764 | 0.0018 | 39.18 | 0.70 | 6/10/84 | 260 |
| ATM-109-1600-02 | 8/17/23 | 0.01240 | 0.00035 | 0.9624 | 0.0035 | 0.1770 | 0.0018 | 41.19 | 0.47 | 6/7/82 | 170 |
| ATM-109-3200-01 | 8/18/23 | 0.01358 | 0.00060 | 0.9656 | 0.0033 | 0.1773 | 0.0018 | 39.81 | 0.71 | 10/27/83 | 260 |
| ATM-109_3-1 | 11/6/21 | 0.01462 | 0.00069 | 0.9711 | 0.0022 | 0.1777 | 0.0018 | 38.70 | 0.74 | 2/23/83 | 270 |
| ATM-109_3-2 | 12/22/21 | 0.01763 | 0.00090 | 0.9701 | 0.0030 | 0.1776 | 0.0018 | 35.79 | 0.80 | 3/9/86 | 290 |
| ATM-109_3-3 | 1/21/22 | 0.01303 | 0.00046 | 0.9718 | 0.0023 | 0.1777 | 0.0018 | 40.48 | 0.57 | 7/27/80 | 210 |
| BR3 total dissolutions | | | | | | | | | | | |
| G14 | 8/12/20 | 0.01306 | 0.00022 | 1.07343 | 0.00069 | 0.1832 | 0.0018 | 40.92 | 0.31 | 9/12/79 | 110 |
| G16 | 7/27/20 | 0.01299 | 0.00053 | 1.0489 | 0.0019 | 0.1820 | 0.0018 | 40.90 | 0.65 | 9/2/79 | 240 |
| G18 | 8/4/20 | 0.01260 | 0.00022 | 1.0591 | 0.0020 | 0.1825 | 0.0018 | 41.42 | 0.31 | 3/6/79 | 110 |
| G21 | 8/6/20 | 0.01335 | 0.00014 | 1.0768 | 0.0010 | 0.1834 | 0.0018 | 40.59 | 0.23 | 1/4/80 | 83 |

Notes:
Only initial $^{85}$Kr/($^{83}$Kr+$^{84}$Kr) ratios and ages calculated from stable Xe isotope ratios are shown. Corresponding calculations based on stable Kr isotopes are not included.

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
