# Peer review of "Krypton-85 chronometry of spent nuclear fuel"

_Geochronology, 2024_

## Referee Comment (RC1)

Review for «Krypton-85 chronometry of spent nuclear fuel" by Balco et al

After having carefully read the paper and checked a few details, I recommend publication but only after some adjustments and improvements.

The more serious points are mentioned below, especially point 5 is relevant as this point is very important for the entire paper.

1. Page 2, line 31: There is the problem with noble gas loss during operation. The authors mention thermally activated diffusion either out of the fuel element or into a bubble. In either way, this process will fractionate the gas. The light isotopes will diffuse faster and will therefore be lost more readily than the heavy isotopes. This process likely needs to be considered but at least should be discussed.
2. Page 5, sample acquisition
   - Both sample are not perfect for determining the age, right? For sample BR3, there was 2 years of irradiation, the one year pause and another year of irradiation. In this one year time gap, some of the 85Kr decays already. In such a scenario, what exactly are you dating? Is the decay during the gap included. If you know the irradiation conditions, time dependent flux, you can correct for this. I think this should be discussed.
   - Sample ATM-109 is, in my opinion, even worse. With the long irradiation time, you start seeing saturation effects in 85Kr. How are they included. My understanding was, that this procedure holds for short irradiation times, i.e., when the increase for stable and radioactive isotopes are more or less linear but that saturation effects are not included. For example, in Fig. 2, the ratio 85Kr / (83Kr + 84Kr) starts decreasing for long irradiation times. How is this treated?
3. I am not too happy with the blank treatment. From our experience, the blank in a cold environment, i.e., not shooting the laser is typically much lower than the blank in a hot environment, i.e., shooting the laser. This has something to do with production and condensation of hot gas and is actually understandable, heating something up produces gas and some material condenses somewhere and thereby produces blank.
4. In addition, there was no complete noble gas extraction. Usually, this also produces fractionation of the released but also of the remaining gas. Has this been tested? Are the noble gas isotope ratios released in the individual steps consistent? This needs a little more work.
5. Page 11: There is some point I cannot understand. First, how can gas loss affect the age? Assuming production is still in the linear range, i.e., short irradiation time, loss of noble gases would affect all isotopes simultaneously and would therefore have no effect on the age, right? This is true for any type of gas losses (location) but in my opinion also true for the time when the gas loss occurred (early or late in the irradiation). The same is true with a change in production rates. If the change is the same of all isotopes, you won't see it in your procedure. For a long irradiation, when the increase of [85]Kr is lower than linear due to saturation effects.

One still loses the same amount of $^{83}$Kr, $^{84}$Kr, and $^{85}$Kr in that hypothetical loss event but now one loses relatively more $^{83}$Kr and $^{84}$Kr relative to $^{85}$Kr

[revised manuscript text omitted]

---

## Author Response (AR1)

Dr. Aeschbach:

Here we summarize revisions to 'Krypton-85 chronometry of spent nuclear fuel'. Our responses to the reviews each included a list of our proposed revisions, and below we index them to their locations in the revised manuscript. In addition, we made minor changes to simplify some potentially confusing aspects of the paper, in particular removing unnecessary discussion of mixed fast and thermal neutron production. Note that the line numbers refer to the revised manuscript, NOT the latexdiff comparison of the original and revised versions.

-- Greg Balco on behalf of all co-authors

**RC1 (Leya):**

1. Add text briefly describing the fact that we did not observe diffusive fractionation during heating.
        -- We added discussion of this in lines 193-202.

2. Add discussion to clarify the process by which gas loss affects Kr and Xe isotope ratios. Specifically, we will clearly distinguish the concept of potential fractionation of gas during diffusive loss from the concept of preferential retention of gas produced later in the irradiation.
        -- We revised lines 229-262 to include additional discussion of this issue.

3. Add discussion to clearly define the concept of an "apparent age" that falls within a period of irradiation, but does not correspond to a specific event.
        -- We added this in lines 98-110.

4. Expand the description of blank corrections as noted above.
        -- We expanded the relevant part of the methods section in lines 185-193.

5. Make various technical corrections and clarify certain areas noted by the reviewer.
        -- These corrections have been made throughout the text.

**RC2 (anonymous):**

1. Add a modest amount of additional context to the discussion of nuclear forensic applications in the abstract and introduction.
        -- We expanded the introduction (lines 9-41), including additional background references, and expanded the abstract.

**CC1 (Popov):**

1. Try to add additional background information on various nuclear processes that are not commonly used in geochronology
        -- We added a number of clarifying sentences throughout the paper.

2. Include equations for initial ratio and age
        -- These now appear in lines 91-96.

3. Add collection details for images
        --  We added this information to Figs. 3 and 4.